# How long is long? Word length effects in reading correspond to minimal graphemic units: An MEG study in Bangla

**Swarnendu Moitra** [1]*, **Dustin A. Chacón** [2,3], **Linnaea Stockall** [1]

**1** Department of Linguistics, Queen Mary University of London, London, United Kingdom, **2** Department of Linguistics, University of Georgia, Athens, Georgia, United States of America, **3** Neuroscience of Language Lab, New York University Abu Dhabi, Abu Dhabi, United Arab Emirates

☯ These authors contributed equally to this work.
* s.moitra@qmul.ac.uk, swarnendu.moitra@gmail.com

**Data Availability Statement:** All data, code, and materials are openly available at: https://osf.io/ptf63/ DOI 10.17605/OSF.IO/PTF63.

**Funding:** This work was supported by Economic and Social Research Council: [ES/V000012/1] to L. Stockall. (https://www.ukri.org/councils/esrc/) and

## Abstract

This paper presents a magnetoencephalography (MEG) study on reading in Bangla, an east Indo-Aryan language predominantly written in an abugida script. The study aims to uncover how visual stimuli are processed and mapped onto abstract linguistic representations in the brain. Specifically, we investigate the neural responses that correspond to word length in Bangla, a language with a unique orthography that introduces multiple ways to measure word length. Our results show that MEG signals localised in the anterior left fusiform gyrus, at around 130ms, are highly correlated with word length when measured in terms of the number of minimal graphemic units in the word rather than independent graphemic units (*akśar*) or phonemes. Our findings suggest that minimal graphemic units could serve as a suitable metric for measuring word length in non-alphabetic orthographies such as Bangla.

## Introduction

In processing visual stimuli, the first brain responses detectable by non-invasive recording occur approximately 100–200ms after presentation. For instance, magnetoencephalography (MEG) recordings show that activity from left fusiform gyrus distinguishes noisy vs. clean visual stimuli at ∼ 100ms (M100), word length effects around ∼ 130ms (M130), and words vs. non-linguistic symbols at ∼ 170ms [1]. In reading, these early brain responses are modulated by psychophysical variables, such as colour attributes and motion features of the visual stimulus [2–4]. However, some findings demonstrate that these earliest brain responses reflect more abstract linguistic analysis, beyond simple psychophysical responses [5]. Word length, measured in number of characters, modulates neural responses ∼ 100-130ms after visual presentation [6–8], and is distinct from brain responses indexing other lexical access processes [9]. Quantifying word length as the number of characters may be a suitable metric for alphabetic writing systems, such as English or Finnish, in which one character generally corresponds to one phoneme, and can surface independently. In other writing systems, corresponding brain responses could reflect either 'lower-level' psychophysical variables, e.g., pixel density or visual

NYU Abu Dhabi Institute under Grant G1001 to A. Marantz. The funders had no role in study design, data collection and analysis, decision to publish, or preparation of the manuscript.

**Competing interests:** The authors have declared that no competing interests exist.

angle subtended, or the initial stages of more complex linguistic analysis, e.g., stages of phonological analysis. Here, we present a magnetoencephalography (MEG) study on reading in Bangla (Bengali), which uses a non-alphabetic orthography in which independent graphemic units (*akśar*, or *ôkkhôr*) are composed of smaller graphemic units, and may correspond to multiple phonemes. This allows for multiple, separable metrics of word length. We find that, in Bangla, MEG signals localising to anterior left fusiform gyrus at ∼130ms correlate best with word length, estimated as the number of minimal graphemic units in the word, rather than *akśar* or phonemes. However, *post hoc* analyses suggest that this brain response may also differentially reflect some phonemic analysis. This finding may help clarify the cognitive mechanism by which visual stimuli map onto more abstract linguistic representations, and the function of the 'visual word form area' more broadly [10].

## Bangla orthography

Bangla (Bengali) is an east Indo-Aryan language spoken predominantly in the South Asian region of Bengal, including Bangladesh and the Indian states of West Bengal, Jharkhand, Tripura, and Assam. Bangla is primarily written in the Bangla or *pūrbônagôri* (পূর্বনাগরী, 'eastern city') script, also shared with several other languages. The Bangla script is an *abugida*. Traditionally, *abugidas* are characterised as writing systems in which consonant characters carry an 'inherent' vowel, in Bangla /ô/ (realised variably as [ɔ] or [o]). Consonants that are followed by other vowels are modified by the addition of a diacritic before, after, under, or around the consonant to 'overwrite' the inherent vowel, e.g., ক /kô/ carries the inherent vowel /ô/, but কী /ki/ কু /ku/ কে /ke/ কো /ko/ are modified with other vowel characters. Consonant sequences are often written as 'conjunct characters', in which elements of both consonants are conjoined into a single, complex character. For instance, ক /kô/ and ল /lô/ can form the conjuncts ক্ল /klô/, ল্ক /lkô/, ক্ক /kkô/, and ল্ল /llô/. We use *character* to refer to the minimal, dependent units of a Bangla symbol, and we borrow the native word *akśar* (Bangla: অক্ষর /ôkkhôr/ 'letter') to correspond to the independent symbol, which may consist of one or more consonant characters, and zero or one vowel characters. Note that the *akśar* is the unit Bangla speakers consider as a 'letter' for the purposes of spelling, dictionary order, etc.

Additionally, the rules mapping Bangla orthography to phonology are complex. As mentioned, the 'inherent vowel' /ô/ is unwritten; it is inferred from the lack of a vowel diacritic after a consonant. However, not all consonants that lack a vowel diacritic are pronounced with the inherent vowel, e.g., the inherent vowel appears after both consonants ব <b> and ড় <r> in বড় /bôṛô/ 'big', although the inherent vowel /ô/ only appears after the first consonant ক <k> but not the second ম <m> in কম /kôm/ 'less'. Bangla contains some digraphs with pronunciations determined by surrounding context, e.g., যা <ya> can correspond to the phoneme /ê/ (realised as [æ] or [e]) or /ja/. Finally, Bangla orthography is conservative, and maintains some orthographic conjunct characters in clusters that have been phonologically simplified, e.g., ম্ব <mb> may be pronounced as /mô/ or /mbô/, depending on the surrounding context. Taken together, the number of phonemes does not perfectly correspond to the number of characters or *akśar* in the word. This property of Bangla allows us to disentangle different metrics of word length—graphemes, *akśar*, and phonemes, which may correspond to different processing strategies for reading Bangla. These properties are exemplified up in Fig 1A with the Bangla word ক্লান্ত klantô 'tired', which consists of 6 phonemes (/klantô/), 5 characters, and 2 *akśar*.

Here, we report on an MEG study in Bangla to clarify the well-attested effect of word length on early neural responses (M100, M130). We leverage the imperfect correlations between three methods of reckoning word length to better clarify what dimension of 'word length'

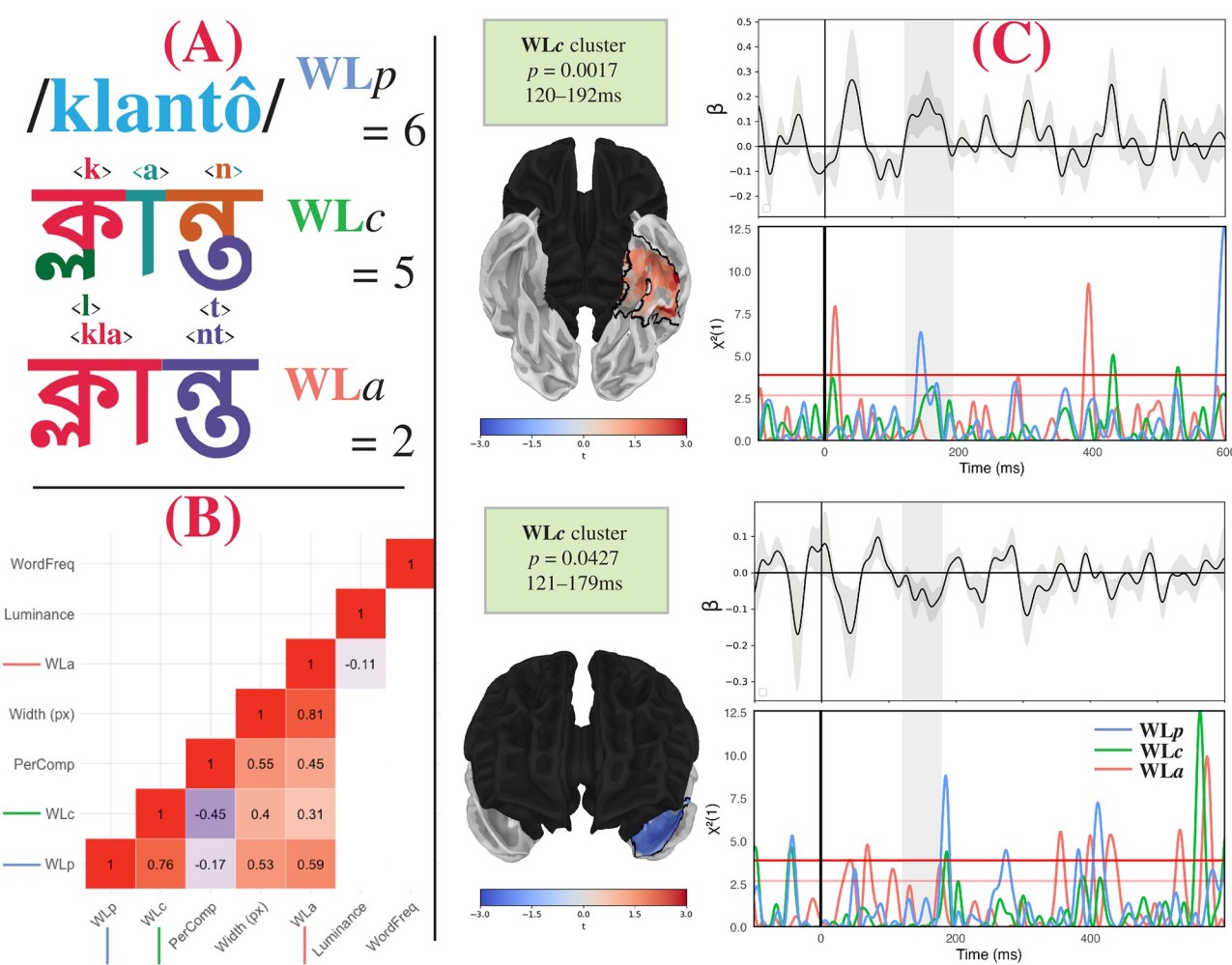

**Fig 1.** Illustration of A: Word length measures, B: Correlation matrix and C: Results. A: Illustration of the separation of word length as measured in phonemes, characters, and *akśar* with the word ক্লান্ত klantô 'tired'. This word consists of 6 phonemes, 5 characters (one for each phoneme, excluding the inherent vowel /ô/), and 2 *akśar* (one for the conjunct consonant + vowel combination ক্লা<kla> and one for the conjunct consonant combination ন্ত <nt>). B: Correlation matrix of the regressors in the experiment. This consists of WordFreq (word frequency, in parts-per-million), luminance, $WL_a$ (word length in *akśar*), PerComp (perimetric complexity, normalised by dividing by $WL_c$), $WL_c$ (word length in characters), and $WL_p$ (word length in phonemes). Non-significant correlations are absent; shading corresponds to magnitude of correlation coefficient. C: Results of two-stage regression analysis and *post hoc* likelihood ratio tests [11]. Both clusters correspond to $WL_c$ beta coefficients, with corrected *p*-values and time windows given in the inset boxes. Each brain plot demonstrates the spatial coordinates of the significant clusters outlined in black. Red and blue shading correspond to positive and negative *t*-values of the second stage regression, in the peak of the cluster timecourse. Timecourse plots show the beta coefficients of the cluster (top timecourse) and the results of the likelihood ratio tests for the three definitions of word length (bottom timecourse). Gray shading indicates the temporal extent of the cluster. Red lines in bottom timecourses correspond to $\chi^2(1) = 3.9 \approx p = 0.05$ and $\chi^2(1) = 2.8 \approx p = 0.10$.

early brain responses correspond to. Our analysis also included word frequency, and we included factors corresponding to 'lower level' psychophysical variables as nuisance regressors —including word width, normalised perimetric complexity, and luminance. Word width, the physical width of the visual stimulus, was highly correlated with all three measures of word length and normalised perimetric complexity. Normalised perimetric complexity correlated with word length measured in *akśar*, and negatively correlated with the other two measures of word length. Luminance negatively correlated with word length measured in *akśar*. Finally, word length did not correlate with any other variable of interest (see Fig 1B). We find that brain activity localising to anterior left fusiform gyrus significantly correlates with word length,

as measured by characters, or the minimal graphemic units constituting a Bangla character. *Post hoc* analyses on the brain activity in these spatial coordinates suggest that this effect may also reflect some initial stages of some phonological processing. The number of *akśar* did not produce significant correlations with brain responses, suggesting that this traditional reckoning is not a crucial component for the early stages of reading in the brain. Our analysis revealed no significant sensitivity to word frequency in the early stages, as observed in EEG [9].

## Materials and methods

### Participants

Twenty-four right-handed, self-reported native speakers of Bangla with normal or corrected-to-normal vision participated in the study. They were recruited from the New York University and surrounding communities in Abu Dhabi. Language history in the form of a questionnaire was collected to screen for eligibility, and written informed consent was provided by all participants prior to the experiment. Compensation was provided upon completion. The NYU Abu Dhabi Institutional Review Board approved all experimental protocols. The recruitment took place between 2/03/2022 to 15/11/2022.

### Materials

Our study consisted of 304 morphologically complex words: 152 grammatical and 152 pseudo-words. Grammatical words were existing Bangla words. The grammatical words were either abstract nouns beginning with the derivational prefixes প্রতি *prôti-* and দুঃ *duḥ-*, noun-noun compounds (জলকামান *jôl-kaman* 'water-cannon'), or concrete nouns with classifier, case, or number suffixes (classifier টি -ṭi; plural classifiers গুলো *-gulo*, গুলি *-guli*; human classifier জন *-jôn*; accusative case কে *-ke*; locative case য়/এ *-e*; classifier+genitive case টির -ṭi-r, টার, -ṭa-r; plural genitive/dative দের *-der*). The pseudowords were created with the prefixes প্রতি *prôti-* and দুঃ *duḥ-*, but attached to morphologically unlicensed stems to produce unattested words. Importantly, our pseudowords are composed of familiar Bangla prefixes and stems, are not immediately detectable as non-words and are orthographically and phonologically well-formed. Their status as ungrammatical words is only detectable after morphological analysis following word-form identification. The stimuli were constructed to address an unrelated question related to processing of multimorphemic words.

### Procedure

All visual stimuli were presented in the centre of the screen. A fixation cross appeared pre-stimulus for 500ms, followed by the word, which stayed on the screen for 2000ms. The inter-stimulus interval (ISI) was jittered, varying between 300-500ms. Stimuli were presented in black Vrinda Bangla font in Bangla text (font size 25) against a gray background using the experiment control Presentation$^{®}$ software (Version 23.0, Neurobehavioral Systems, Inc., Berkeley, CA, www.neurobs.com). The stimuli were projected onto a screen inside the Magnetically Shielded Room (MSR; Vacuumschmelze, Hanau, Germany) using a projector. The stimulus was equally divided into two blocks. Each block's stimulus order was completely randomised, and the order of presentation of the blocks alternated between participants.

### Task

The experiment consisted of a lexical decision task in which participants were presented with strings of characters appearing in the middle of the screen. Participants were instructed to indicate via button press with the non-dominant (left) hand whether they recognised the string

as a word of their language, and to answer as quickly and accurately as possible. The buttons were counterbalanced; half of the participants indicated yes by pressing the left button on the response box and the other half by pressing the right button. Between the blocks, participants could take a self-timed break to perform small movements to remain comfortable. The average total time for the experiment was 15 minutes.

## Data acquisition

The head contours of all participants were digitised using a hand-held FastSCAN laser scanner (Polhemus, VT, USA) to enable coregistration during data preprocessing; five fiducial points were marked (left and right preauricular points, and three spots on the forehead) on each participant's head and were also digitised using the scanner. While inside the MSR, marker coils were placed on the points to localise the participant's head with respect to the MEG sensors. We took marker measurements before and after the experiment to record the relative movement of the participant's head, which was corrected during coregistration. Continuous recording of MEG data was performed using a 208-channel axial gradiometer system (Kanazawa Institute of Technology, Kanazawa, Japan) at a sampling rate of 1000 Hz. An online low-pass filter with a cutoff frequency of 200 Hz and a high-pass filter with a cutoff frequency of 0.01 Hz was applied on-line.

## Data analysis

### Preprocessing

Three reference channels positioned at a distance from the head of the participant recorded noise inside the MSR. To reduce MEG noise, we used the Continuously Adjusted Least Squares Method (CALM) [12] in MEG160 (Yokogawa Electric Corporation and Eagle Technology Corporation in Tokyo, Japan) for each participant. Next, we imported the denoised data into MNE-Python [13] and applied band-pass filtering between 1 and 40 Hz. Subsequently, we removed bad channels using visual inspection, and then extracted epochs without applying a baseline from 100 ms before the stimulus onset to 600 ms post-stimulus. We used Independent Component Analysis (ICA) to remove noise sources. We set 95% as the cutoff percentage of signal variance to be captured by the components identified by ICA. In order to avoid random noise bursts from contributing to explaining the variance, a first round of ICA computations was performed, in which we iteratively excluded epochs with values that were outliers compared to other epochs in the resulting components, for the sake of calculating a subsequent cleaner ICA. For each component, we defined the set of outlier epochs conservatively as the smaller of two sets: the set of epochs whose variance levels were in the top 5% of all epochs and the set of epochs whose maximum values exceeded the third-quartile of all epochs' maximum values by three times the inter-quartile range. Because the exclusion criteria were conservative, most components did not contribute to any epoch exclusions, and the algorithm targeted primarily those epochs which contributed to significant noise bursts in one or more components. Subsequently, we re-calculated the ICA solution only on the basis of the remaining epochs and removed those ICA components belonging to identifiable sources (heartbeats, blinking). The remaining components constituted the cleaned mapping from MEG channels to the ICA solution; this mapping was then applied in reverse to all of our epochs (including those excluded to calculate the ICA solution), producing an ICA-cleaned signal. Following this, bad channels were first interpolated, and then a 100ms baseline correction was applied. Epochs exceeding the absolute threshold value of 2 pT were rejected (0—15 (min-max) and 2% of epochs removed, across participants), resulting in the final cleaned sensor-space signal.

## Source reconstruction

Sensor data was projected into source space using MNE-Python. MEG data were coregistered with the FreeSurfer template brain 'fsaverage' (CorTechs Labs Inc., California, USA and 175 MGH/HMS/MIT Athinoula A. Martinos Center for Biomedical Imaging, Massachusetts, USA). The fsaverage template brain was scaled to match the digitised headshape and fiducial markers. An ico-4 source space was then built with 2562 vertices per hemisphere using the minimum norm estimate algorithm [14]. A forward solution was then computed using the boundary element method (BEM) [15]. Channel noise covariance matrices were estimated using the baseline period (100ms before each epoch) and regularised using the automatic method [16]. Combining the forward solution and the noise covariance matrix, an inverse solution was computed for each evoked response. For computing the inverse solution, we used a fixed dipole orientation. This method places a dipole orthogonal to the cortical surface at each source point. This allows for estimating signed (i.e., exiting or entering) signals, which provides an additional dimension for detecting differences in neural activity than free dipole orientation [17]. Finally, the signals were noise normalised in the spatial dimension, yielding a dynamic statistical parameter map (dSPM) [18]. Subsequent analyses were conducted on single-trial spatio-temporal timecourse (STCs).

## Source-space analysis

Because our research question involves initial stages of processing word forms, prior to lexical access, our analysis includes both grammatical and ungrammatical words. Preliminary analyses on only grammatical words did not produce any reliable effects, likely due to weak power. Because we are focused on the initial stages of interpreting orthographic features, before morphological analysis occurs [5, 19], there are no expected differences between grammatical and pseudoword stimuli.

For analysis, we conducted a two-stage spatio-temporal cluster-based permutation test. For each individual, we fit a regression to each time point and source point, across all trials. This produced a beta estimate for each factor of the regression, at each time and source point for the individual. Afterwards, one-tail $t$-tests were conducted on the beta values, determining which beta values were significantly different than zero. In the second stage, spatio-temporal cluster-based permutation tests were conducted on the $t$-values, following the procedure described by [20]. Clusters were formed by identifying significant ($p < 0.05$) $t$-values that were adjacent in space and time that had the same sign (i.e., positive or negative). The cluster algorithm was further constrained such that the minimum cluster contained 20 source points and 20ms. Cluster statistics were calculated by summing the $t$-values. The null distribution was then estimated by randomly permuting the labels and recomputing cluster statistics 10,000 times. Corrected cluster-level $p$-values were estimated by comparing the original test statistic against this bootstrapped null distribution, and were considered significant at $\alpha = 0.05$. Search parameters for the clustering algorithm were constrained to bilateral temporal and occipital cortices, from 70–210ms. Time points were selected by visual inspection of raw sensor data, and were selected to include expected M100, M130, and M170 responses e.g., [19]. Both grammatical and pseudoword trials were included, and trials were not excluded based on participants' responses. Regression factors included three variables of word length, as defined by the number of *akśar* ($WL_a$), the number of characters ($WL_c$), and the number of phonemes ($WL_p$). We also included a factor of whole-word frequency, since this may effect neural responses at later time windows. Finally, we included psychophysical variables corresponding to the width of word in pixels, the estimated luminance of the word printed against the gray background, and the normalised perimetric complexity [21]. Perimetric complexity is the

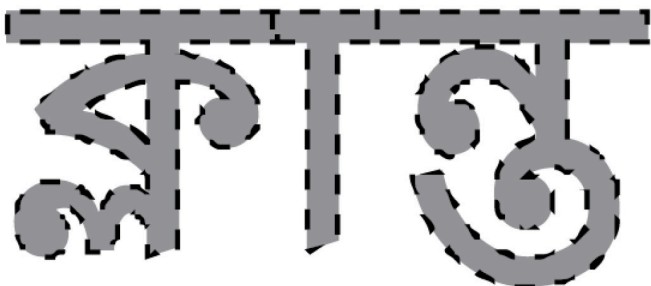

**Fig 2. Illustration of perimetric complexity.** Perimetric complexity is the square of the length of the perimeter of a word (dashed line) divided by the total ink area (light gray). This ratio is illustrated here for the word ক্লান্ত klantô 'tired'. Our models normalise perimetric complexity by dividing perimetric complexity by $WL_c$.

square of the length of the perimeter divided by the total ink area, Fig 2. Perimetric complexity is demonstrated to be near proportional to identification of letters, across fonts and writing systems. We divide the total perimetric complexity by the number of characters ($WL_c$), thus deriving a normalised estimate of perimetric complexity by character. This normalisation is suggested by [9], as a way to deconfound perimetric complexity with the length of a word. These psychophysical variables were included because they could plausibly effect the earlier (M100) response in earlier visual cortices. The full regression model in the two-stage regression model is given in Eq (1).

$$dSPM \sim 1 + WL_a + WL_c + WL_p + Luminance + Width_{px} + \frac{Perim.Complex}{WL_c} + WordFreq_{PPM} \quad (1)$$

Word length metrics were calculated by using the native word-length functions in Microsoft Excel and Python, which estimate $WL_a$ and $WL_c$ respectively. In calculating $WL_c$, we further 'penalised' conjunct characters by including the combining character hasant ্, which is used to render complex conjunct consonant characters. $WL_p$ was calculated using the Bangladesh Bangla to IPA transcription tool (http://www.ipa.bangla.gov.bd/) to represent each word as a sequence of international phonetic alphabet (IPA) characters. IPA diacritics corresponding to aspirated stops ($^h$) and dental articulation ( ̪ ) were removed before counting, since these diacritics resulted in all aspirated and dental stops being counted twice. Additional ad hoc corrections were made for consistency in transcription.

The three definitions of length were expected to be highly inter-correlated—longer words should be longer by all three definitions of word length. We similarly expected some correlation between the psychophysical variables and the word length metrics. The correlation matrix of these variables is shown in Fig 1B. We do not orthogonalize our collinear variables, as we are specifically interested in understanding the effect of the word length variables, regressing out the psychophysical responses [22]. We address these multicollinearities in the *post hoc* analyses. We also included lexical frequency as a regressor. Lexical frequency was estimated by searching the Bangla IndicNLP corpus [23] for all words that contain the experimental items. Word frequency was smoothed by adding 0.1 to all counts. Non-word stimuli from the experiment therefore had a total raw count of 0.1. Frequency was represented as 'parts-per-million' (PPM) in the regression model. Psychophysical variables were estimated by rendering each Bangla word in the Vrinda font as a monochromatic image with aliasing against a gray background, using the Python library PIL [24]. Luminance and word width were measured in-

software by rendering the word as an image and using in-software functions. These measures are proportional to the relevant physical variables of interest as projected on the screen. Word width as measured in pixels is used as a proxy for visual angle subtended on the screen. Perimetric complexity was calculated following the procedure described by [21]. However, this measure is highly correlated with word length as measured in characters; thus, we normalised perimetric complexity by dividing it by $WL_c$ (see also [9]). We also computed the height of each word in pixels, and the number of pixels contained in the area of the word. However, these variables were highly correlated with other psychophysical variables, and thus were excluded from further analysis.

We ran a separate two-stage spatio-temporal cluster-based permutation to determine whether the MEG signal distinguished between grammatical words and pseudowords within the time and search coordinates of interest. This analysis was conducted with the same parameters as the analysis of interest, except with a single factorial regressor, coding each trial as a grammatical word or a pseudoword.

## Results

### Behavioural results

A linear mixed-effects model fit to the RT results show (see Fig 3) that participants were slower to respond to pseudowords compared to grammatical words ($\beta$ = 65.84, SE = 11.47, $t$ = 5.74, $p < 0.001$). A logit mixed-effects models fit to participants' acceptance/rejection responses show that grammatical words were accepted more often than pseudowords ($\beta$ = -3.70, SE = 0.12, $z$ = -30.18, $p < 0.001$). Model results are reported in Table 1. All statistical analyses were performed in R (v4.3.2) [25] using the lme4 package [26]. Button presses reflect the output of participants' interpretation of a word after lexical access and morphological analysis, whereas we focus on the MEG signal correlating to the pre-morphological orthographic processes here.

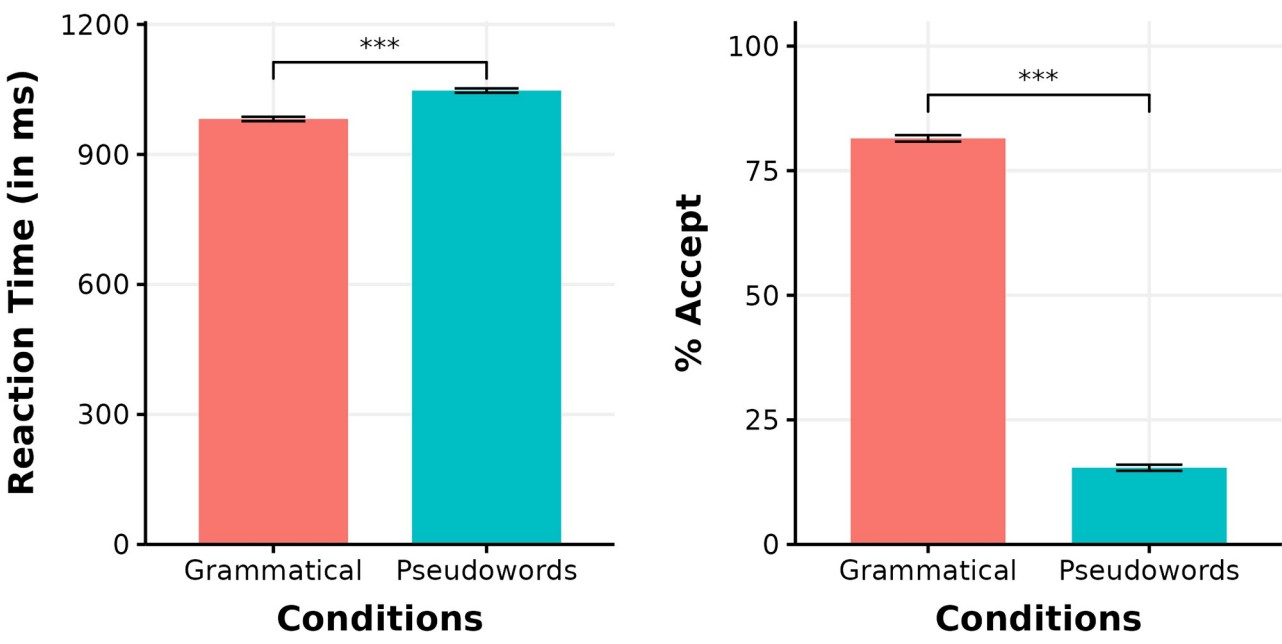

**Fig 3. Behavioral results.** Mean reaction time (in milliseconds) and percentage acceptance across both conditions.

**Table 1. Model summary.**

| | RT[1] | | | | |
|---|---|---|---|---|---|
| Predictors | $\beta$ | SE | CI | t value | p value |
| (Intercept) | 986.27 | 32.23 | [923.10, 1049.45] | 30.61 | **<0.001** |
| Condition [Pseudowords] | 65.84 | 11.47 | [43.36, 88.33] | 5.74 | **<0.001** |
| | Response[2] | | | | |
| Predictors | $\beta$ | SE | CI | z value | p value |
| (Intercept) | 1.79 | 0.13 | [1.54, 2.05] | 14.15 | **<0.001** |
| Condition [Pseudowords] | -3.70 | 0.12 | [-3.95, -3.46] | -30.18 | **<0.001** |
| Formula: | [1] RT $\sim$ Condition + (1|Participant) + (1|Item) | | | | |
| | [2] Response $\sim$ Condition + (1|Participant) + (1|Item) | | | | |

Summaries of the linear mixed effect model for reaction time (RT) and the logistic mixed effect model for Response.

## MEG results

In the main analysis, we identified three significant clusters. Two clusters corresponding to $WL_c$ were identified. The first cluster was identified in anterior left fusiform gyrus, from 120–192ms ($p = 0.0017$). This cluster consisted of positive $t$-values, corresponding to a positive correlation between word length in characters and MEG activity (dSPM). The second cluster was identified in the left temporal pole, from 121–179ms ($p = 0.0427$). In contrast, this cluster consisted of negative $t$-values, corresponding to a negative correlation between word length in characters and MEG activity. No significant clusters of $WL_p$ or $WL_a$ were identified in this analysis. The spatial extent and time course of beta coefficients are plotted in Fig 1C. These two clusters suggest that the brain responses peaking around 130–150ms (M130) correlate primarily with the length of a word as expressed in characters. The spatial distribution of the positive cluster in anterior left fusiform gyrus appears to be consistent with previous findings on early stages of reading in the brain [1, 17, 27], which localises this brain activity to the 'visual word form area'. The timing of these results is also consistent with EEG findings, which show a profile for word length effects between 100–150ms that is distinct from other early lexical access effects, such as word frequency and concreteness [9].

We also found a significant cluster of normalised perimetric complexity. Although this was included as a nuisance regressor, we report on it here since it overlaps with the $WL_c$ cluster. This cluster localised to left fusiform gyrus, from 151-180ms ($p = 0.0189$), and showed a positive correlation between normalised perimetric complexity and MEG signals. This is shown in Fig 4. No significant effects of any of the other variables was observed.

Because of the multicollinearity of our regressors, we also conducted a *post hoc* analysis on the spatial coordinates of the cluster. We averaged the raw activation from the spatial coordinates of both $WL_c$ clusters for each participant and trial. Then, for each millisecond, we fit a linear model consisting of the same form as the two stage regression in (1), fit over the averaged timecourses extracted for each participant and trial. For each word length variable, we then fit a separate reduced model, removing the word length variable, at each millisecond. We then conducted a likelihood ratio test at each time point comparing each model, to determine whether the more complex model incorporating the word length variable is a better fit than the model without the word length variable. This then produces a time-course of $\chi^2(1)$ test-statistic for each word length variable, showing when each factor contributes significantly to explaining the data. These distributions are given in Fig 1C.

This *post hoc* analysis reveals a somewhat different pattern. In the spatial coordinates of the cluster located in anterior left fusiform gyrus, the likelihood ratio tests suggest that $WL_p$ may

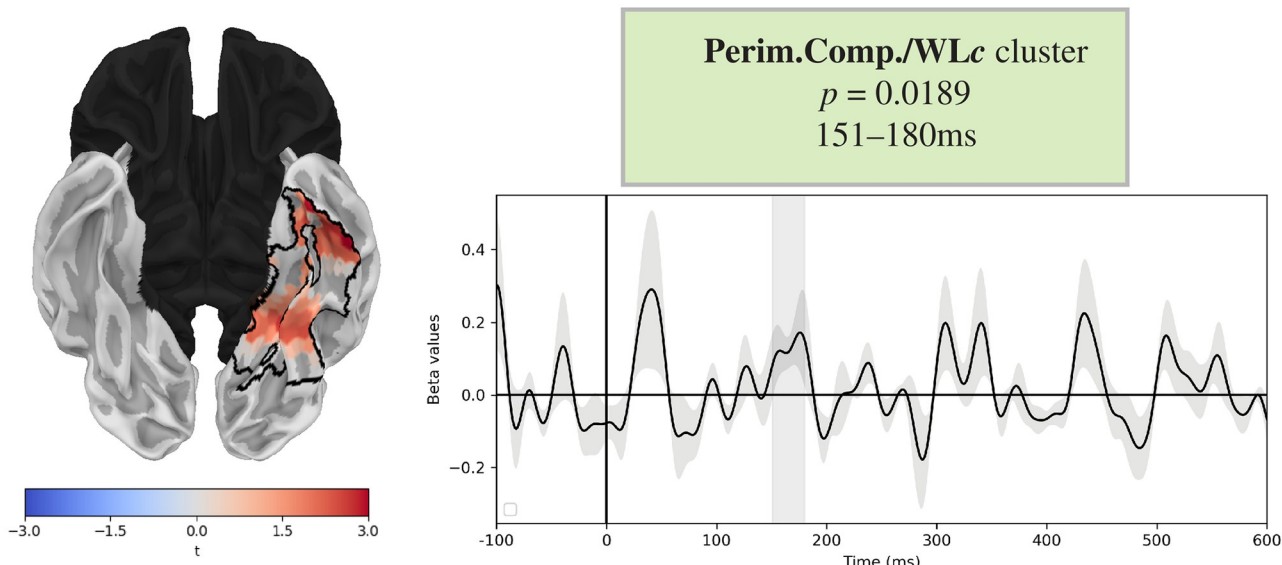

**Fig 4. Normalised perimetric complexity cluster.** Results of two-stage regression analysis. Cluster corresponds to normalised perimetric complexity, or the square length of the perimeter of the word divided by the ink area internal to word, divided by the word length in minimal graphemic units, $WL_c$. Corrected *p*-value and temporal extent given in the green box. Brain plot demonstrates the spatial coordinates of the significant clusters outlined in black. Red shading corresponds to the positive *t*-values of the second stage regression, in the peak of the cluster timecourse. Timecourse plots show the beta coefficients of the cluster, and gray shading indicates the temporal extent.

contribute significantly to explaining the data during the critical time window (120–192ms), as seen by the peak of $\chi^2$-values surpassing the $\chi^2(1) = 3.9 \approx p = 0.05$ threshold. This effect was likely not identified in the two-stage regression analysis because we constrained the clustering algorithm to a minimum time-window of 20ms, and this peak lasted only 12ms. The $WL_c$ timecourse, by comparison, also briefly peaked above the $\chi^2(1) = 2.8 \approx p = 0.10$ threshold. By contrast, the likelihood ratio tests in the negative cluster located in temporal pole did not suggest any significant contribution of any word length variable. Additionally, there did not appear to be any significant contribution in either cluster of $WL_a$.

The two-stage regression model fit to the factorial variable Grammatical vs. Pseudoword did not identify any significant clusters with the search and clustering parameters reported here. This is compatible with the expectation that morphological and lexical analysis is likely not launched during the M100 and M130 time windows.

## Discussion

It is well-attested that the earliest brain responses to words in reading correlate with word length. However, 'word length' is highly confounded with psychophysical variables, and can be quantified differently depending on the orthographic and phonological rules of the language. Here, we presented an MEG study in Bangla, a language with an *abugida* writing system, which enables quantifying words in terms of minimal letter-forms (here, 'characters'), and larger, quasi-syllabic units (here, *akśar*). Additionally, because of complex orthography-phonology mapping rules (similar to better studied languages like English and French), these two metrics are not perfectly correlated with the number of phonemes in a word, allowing for disentangling three separate measures of word length.

We found that MEG activity in anterior portions of left fusiform gyrus significantly correlated with word length as quantified by characters, or the minimal orthographic units of a

character. This suggests that the 'visual word form area' in Bangla readers may reflect computations that are sensitive to identifying phonologically-relevant units of a word, rather than coarser-grained units such as the quasi-syllabic *akśar*, or a more abstract, finer-grained analysis in terms of phonemes. Furthermore, this study provides clear evidence that visual word form activity corresponds to properties of the orthography of a written word, rather than psychophysical responses (which are more highly correlated with *akśar* count rather than character count). This can help constrain theories of reading and orthographic processing in the brain, which are mostly developed on data from languages with tighter correlations between visual complexity, word length, and phoneme number. Moreover, *post hoc* analyses in the spatial coordinates of this anterior left fusiform area suggest that word length as measured by the number of phonemes in a word may also exert an effect on visual word form activity. Further research in both Indic *abugidas* with similar psychophysical and orthographic properties to Bangla (e.g., Hindi/Nepali/etc. *Devanāgari*, Punjabi Gurmukhī), other *abugidas* with different orthography/phonology rules (e.g., Tibetan, Thai, Amharic/Ethiopic Ge'ez, Cree Syllabics), and writing systems that involve composition of complex characters from minimal parts (e.g., Korean *hangeul*) may further constrain what kinds of computations left fusiform gyrus contributes to orthographic and lexical processing.

## The function of fusiform in the reading network

The fusiform gyri are part of a broader network of brain regions that facilitate reading, and serve different functions. Other brain areas that are critical for reading involve the 'dorsal pathway', which consists of left temporo-parietal regions, and may facilitate in mapping orthographic form to phonemic representations, and the 'ventral pathway', which consists of left temporal regions and inferior frontal regions, which may facilitate mapping whole word-forms to meanings and detecting 'sight words' [28–30]. These reading strategies may also be deployed differently at different stages of development and across different writing systems [31, 32]. The occipito-temporal regions, including the fusiform gyri, are a crucial component of this network, since they serve as the first gateway mapping the visual stimuli to more abstract representations, which may be used by either reading strategy. During the early stages of reading development, activation in the occipital-temporal regions has been shown to increase with reading skills acquisition, in line with the view of progressive formation of alphabetic and orthographic representations. [33, 34]

Our results contribute to our understanding of the fusiform gyri and their relationship to the broader reading network by demonstrating that fusiform gyri facilitate sub-lexical analysis of the word-form [5, 9]. Additionally, our results show that in an abugida like Bangla, these processes are sensitive to the minimal graphemic units, which roughly correspond to phonemes, rather than larger composed graphemes (*akśar*), which roughly correspond to syllables.

## Acknowledgments

We want to thank Prof. Alec Marantz and Dr. Samantha Wray for their constructive feedback. We would also like to thank Dr. Ishani Guha and Dr. Bidisha Bhattacharjee for their help in preparing the stimuli. We also want to thank Dave Cayado and the New York University Abu Dhabi Neuroscience of Language Lab members for their assistance during the data collection.

## Author Contributions

**Conceptualization:** Swarnendu Moitra, Dustin A. Chacón, Linnaea Stockall.

**Data curation:** Swarnendu Moitra, Dustin A. Chacón.

**Formal analysis:** Swarnendu Moitra, Dustin A. Chacón, Linnaea Stockall.

**Funding acquisition:** Linnaea Stockall.

**Investigation:** Swarnendu Moitra.

**Methodology:** Swarnendu Moitra, Dustin A. Chacón.

**Project administration:** Linnaea Stockall.

**Resources:** Linnaea Stockall.

**Visualization:** Swarnendu Moitra, Dustin A. Chacón, Linnaea Stockall.

**Writing – original draft:** Swarnendu Moitra, Dustin A. Chacón, Linnaea Stockall.

**Writing – review & editing:** Swarnendu Moitra, Dustin A. Chacón, Linnaea Stockall.

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
