## [Decision Letter · Decision Letter 0]

26 Dec 2023

PONE-D-23-30063How long is long? Word length effects in reading correspond to minimal graphemic units: An MEG study in BanglaPLOS ONE

Dear Dr. Moitra,

Thank you for submitting your manuscript to PLOS ONE. After careful consideration, we feel that it has merit but does not fully meet PLOS ONE’s publication criteria as it currently stands. Therefore, we invite you to submit a revised version of the manuscript that addresses the points raised during the review process.

Both reviewers agreed that your manuscript is worthy of publication once you have fully addressed the minor revisions. Each reviewer has provided some clear guidance on how to revise your submission. In particular, please fully address the reviewer concerns regarding the clarity of the method, and the inclusion of the additional analyses.

We look forward to receiving your revised manuscript.

Kind regards,

John Blake, PhD

Academic Editor

PLOS ONE

Reviewers' comments:

Reviewer's Responses to Questions

**Comments to the Author**

1. Is the manuscript technically sound, and do the data support the conclusions?

Reviewer #1: Yes

Reviewer #2: Yes

2. Has the statistical analysis been performed appropriately and rigorously? 

Reviewer #1: Yes

Reviewer #2: Yes

3. Have the authors made all data underlying the findings in their manuscript fully available?

Reviewer #1: Yes

Reviewer #2: Yes

4. Is the manuscript presented in an intelligible fashion and written in standard English?

Reviewer #1: Yes

Reviewer #2: Yes

5. Review Comments to the Author

Reviewer #1: The authors present a study of the neural responses that correspond to word length in Bangla, an Indo-Aryan language which is a branch of the Indo-Iranian language family. Bangla is written in abugida script which is a semi-syllabic writing system where consonant letters generally represent an inherent phonetic vowel which may be suppressed in initial consonant clusters or in final consonants.

I suggest that the authors incorporate the following information to provide more context for their research questions.

The reading network in the left hemisphere includes the temporoparietal cortex, the occipital temporal cortex, and the inferior frontal cortex. The left ventral occipitotemporal cortex is a key region for orthographic processing because the "visual word form area" is located in this region. The left OT cortex begins to show specialization for orthographic processing, as reading acquisition increases.

There are two important pathways that are engaged during reading. In the dorsal reading pathway is the left temporo-parietal area which is involved in phonological representation and phonological assembly. In the ventral reading pathway is the left inferior temporal gyrus which is involved in rapid orthographic recognition and sight word reading.

There are three stages of reading development. During logographic reading, a reader memorizes the meaning of a word based on the shape of the letters and the word. During phonological reading, a reader applies grapheme-phoneme-correspondence to convert scripts to sounds and then accesses meaning. During orthographic reading, a reader recognizes sight words from memory without having to convert graphemes to phonemes.

Reviewer #2: The study's use of magnetoencephalography (MEG) to investigate the impact of word length on brain activity in Bangla, a non-alphabetic script, is insightful. It demonstrates that brain activity in the anterior left fusiform gyrus correlates with word length measured by minimal graphemic units, challenging traditional metrics and theories on early-stage reading sensitivity to word frequency.

However, the methodology requires additional clarity:

- The study should specify if the regression analysis uses the contrast of word and non-word MEG responses or only words. This clarification would enhance methodological robustness.

- If not done, the integration and comparative analysis of responses to both words and pseudowords (non-words) would deepen the understanding of language-specific processing.

- The detailed task performance findings in the analysis are necessary for a comprehensive understanding of the relationship between behavioral responses and MEG data.

Minor,

- Please ensure consistency between the color bar and the localized sources. The colormap of the maps looks 'hot' but the colorbar is blue-white-red.

These revisions would strengthen the study's contribution to cognitive processing research in non-alphabetic languages. I recommend this study for publication in Plos One with the aforementioned revisions.

6. PLOS authors have the option to publish the peer review history of their article (what does this mean?). If published, this will include your full peer review and any attached files.

Reviewer #1: **Yes: **Kyle PerkinsI

Reviewer #2: **Yes: **Vahab Youssof Zadeh

---

## [Author Response · Author response to Decision Letter 0]

13 Feb 2024

Dear Managing Editor, 

Thank you for the consideration of our manuscript 'How long is long? Word length effects in reading correspond to minimal graphemic units: An MEG study in Bangla' for the journal PLoS One. We appreciate the feedback and advice that was provided to us by our reviewers. We've updated the manuscript, implementing most of the suggestions that were provided to us by the reviewers. We believe that this has resulted in a stronger and clearer paper that makes a better, direct contribution to the neuroscience of reading. We have also highlighted the changes, for better comparison with the original submission. 

Below, we've responded to each of the reviewers' comments, clarifying how we've implemented their suggestions to improve the paper. Our comments are provided in red. 

–––––––––––––––––

Reviewer 1

Reviewer #1: The authors present a study of the neural responses that correspond to word length in Bangla, an Indo-Aryan language which is a branch of the Indo-Iranian language family. Bangla is written in abugida script which is a semi-syllabic writing system where consonant letters generally represent an inherent phonetic vowel which may be suppressed in initial consonant clusters or in final consonants.

I suggest that the authors incorporate the following information to provide more context for their research questions.

The reading network in the left hemisphere includes the temporoparietal cortex, the occipital temporal cortex, and the inferior frontal cortex. The left ventral occipitotemporal cortex is a key region for orthographic processing because the "visual word form area" is located in this region. The left OT cortex begins to show specialization for orthographic processing, as reading acquisition increases.

Response : “Thank you for this suggestion. We agree that acknowledging that our research focuses on one earlier component of the reading network is relevant for understanding the broader context of our results. We believe that our findings clarify the function of this brain region and its role in the initial stages of orthographic analysis. We've updated the manuscript with a subsection in the Discussion where we implement these suggestions.”

There are two important pathways that are engaged during reading. In the dorsal reading pathway is the left temporo-parietal area which is involved in phonological representation and phonological assembly. In the ventral reading pathway is the left inferior temporal gyrus which is involved in rapid orthographic recognition and sight word reading.

There are three stages of reading development. During logographic reading, a reader memorizes the meaning of a word based on the shape of the letters and the word. During phonological reading, a reader applies grapheme-phoneme-correspondence to convert scripts to sounds and then accesses meaning. During orthographic reading, a reader recognizes sight words from memory without having to convert graphemes to phonemes.

Response: “Thank you for this suggestion. We've included a discussion of these matters in the same new subsection in the Discussion. We agree that our results may have some significance for better understanding how the brain implements different kinds of reading strategies, and how these reading strategies may differ between readers and writing systems. However, since we are focusing primarily on adult readers of Bangla, we did not focus substantially on the development of readers. We agree that acknowledging the neural substrates of different strategies is important. In our new section, we suggest that our results demonstrate that the first 'way-station' of orthographic processing is sensitive to smaller units that can roughly correspond to phonemes, which may be key to better theorising the reading strategies that readers of abugidas use and the function of fusiform gyri in the broader reading network.”

Reviewer #2: The study's use of magnetoencephalography (MEG) to investigate the impact of word length on brain activity in Bangla, a non-alphabetic script, is insightful. It demonstrates that brain activity in the anterior left fusiform gyrus correlates with word length measured by minimal graphemic units, challenging traditional metrics and theories on early-stage reading sensitivity to word frequency.

Response: “Thank you, we hope our findings are relevant and contribute to our understanding of the fusiform gyrus and its function in the broader reading network.”

However, the methodology requires additional clarity:

- The study should specify if the regression analysis uses the contrast of word and non-word MEG responses or only words. This clarification would enhance methodological robustness.

Response: “Thank you for this point. The original manuscript did not make this point clearly. We include both grammatical words and ungrammatical non-words since we are focusing on the processes relevant to the initial detection of word-forms, before lexical access or morphological analysis. It is important to note that our non-words are not unfamiliar stems (like 'dax') or implausible letter sequences (like 'xkq'), but rather morphologically ungrammatical words that are composed of familiar stems and affixes (like 'relaugh' or 'redog'). Thus, the identification of basic word form properties are not expected to be substantially different between our grammatical words and pseudowords.”

- If not done, the integration and comparative analysis of responses to both words and pseudowords (non-words) would deepen the understanding of language-specific processing.

- The detailed task performance findings in the analysis are necessary for a comprehensive understanding of the relationship between behavioral responses and MEG data.

Response: “The original manuscript contained the response times to words and pseudowords in the lexical decision task. However, we've included a more detailed discussion of the results, including the acceptance rates of the words. As clarified in the new subsection in Results, reaction times were significantly longer for the pseudowords compared to the grammatical words, and the pseudowords were accepted substantially less often than the grammatical words, as expected. This has been included in the Results section.

We also conducted an analysis to compare the MEG responses to Grammatical vs. Pseudowords in our search parameters. As expected, there were no significant clusters distinguishing between these stimulus types, since we are examining the brain responses before morphological or lexical analysis has occurred, in the occipito-temporal regions that are primarily implicated in mapping visual stimuli to linguistic representations, as opposed to processing those representations.”

Minor,

- Please ensure consistency between the color bar and the localized sources. The colormap of the maps looks 'hot' but the colorbar is blue-white-red.

Response: “The original manuscript used the colorbar correctly; as we explained in the caption, the yellow marked the spatial extent of the colour, and the blue-white-red shading indicated the t-values of the cluster at the peak of the timecourse, red corresponding to positive t-values and blue corresponding to negative t-values. We did not average the t-values over time as is typical in plotting the spatial extent of clusters.

However, to avoid this confusion, we've plotted t-values averaged over time, as is more common, and updated the figures accordingly. This has little effect on the basic shape of the clusters that we are plotting”

These revisions would strengthen the study's contribution to cognitive processing research in non-alphabetic languages. I recommend this study for publication in Plos One with the aforementioned revisions.

Response: “We thank the reviewer for the kind and helpful comments!”

---

## [Decision Letter · Decision Letter 1]

21 Feb 2024

How long is long? Word length effects in reading correspond to minimal graphemic units: An MEG study in Bangla

PONE-D-23-30063R1

Dear Dr. Moitra,

We’re pleased to inform you that your manuscript has been judged scientifically suitable for publication and will be formally accepted for publication once it meets all outstanding technical requirements.

Kind regards,

John Blake, PhD

Academic Editor

PLOS ONE

Additional Editor Comments (optional):

Reviewers' comments:

Reviewer's Responses to Questions

**Comments to the Author**

1. If the authors have adequately addressed your comments raised in a previous round of review and you feel that this manuscript is now acceptable for publication, you may indicate that here to bypass the “Comments to the Author” section, enter your conflict of interest statement in the “Confidential to Editor” section, and submit your "Accept" recommendation.

Reviewer #1: All comments have been addressed

Reviewer #2: All comments have been addressed

2. Is the manuscript technically sound, and do the data support the conclusions?

Reviewer #1: Yes

Reviewer #2: Yes

3. Has the statistical analysis been performed appropriately and rigorously? 

Reviewer #1: Yes

Reviewer #2: Yes

4. Have the authors made all data underlying the findings in their manuscript fully available?

Reviewer #1: Yes

Reviewer #2: Yes

5. Is the manuscript presented in an intelligible fashion and written in standard English?

Reviewer #1: Yes

Reviewer #2: Yes

6. Review Comments to the Author

Reviewer #1: I think that the authors have addressed the reviewers' comments and suggestions. I think that the manuscript now meets all the PLOS ONE criteria for publication, and I look forward to seeing the manuscript in print.

Reviewer #2: The authors have successfully addressed my comments and questions. Therefore, I recommend this work for publication.

7. PLOS authors have the option to publish the peer review history of their article (what does this mean?). If published, this will include your full peer review and any attached files.

Reviewer #1: **Yes: **Kyle Perkins

Reviewer #2: **Yes: **Vahab Youssofzadeh

---

## [Editor Report · Acceptance letter]

24 Mar 2024

PONE-D-23-30063R1 

PLOS ONE

Dear Dr. Moitra, 

I'm pleased to inform you that your manuscript has been deemed suitable for publication in PLOS ONE. Congratulations! Your manuscript is now being handed over to our production team.

Kind regards, 

on behalf of

Dr. John Blake 

Academic Editor

PLOS ONE